# OpenReview forum: "Batch size-invariance for policy optimization"
_NeurIPS.cc/2022/Conference — NeurIPS 2022 Accept_

### Official Review · Reviewer_GBX9 · 2022-06-28

**Rating:** 4
**Confidence:** 4
**Soundness:** 3 good
**Presentation:** 3 good
**Contribution:** 3 good

**Summary:**

In the context of reinforcement learning (RL), the paper introduces one property: batch size-invariance and achieving it by decoupling the proximal policy from the behavior policy. The authors use an exponentially-weighted moving average version of learning policy to achieve this property and show improvements on several experiments.

**Questions:**

What is the main difference between gradients accumulation and your methods?

Could you please give some intuitive examples of the necessity of batch size-invariance?


**Limitations:**

In the absence of a more explicit application scenario for their method, it would be better to find a practical application like asynchronous and multi-machine.

**Strengths And Weaknesses:**

Originality: the proposed method is novel. It is somewhat iterative, decoupled policy objective is used in recent works but applies exponentially-weighted moving average for batch size-invariance that has not been noticed before.

Clarity: The paper is well written and clear. The method is soundness and easy to implement. The paper contains many various and multi-level experiments which prove the effectiveness of their method.

Significance: Although the authors mention in the introduction and section 6.2 that using a large batch-size will give a low variance estimation of policy gradient, and it is often prohibited by computational resources such as GPU memory, so we should use their method. But there is a more simple way to achieve this goal like accumulating gradients from multiple back propagation and updating until calculating as many samples as we need. Is there anything I mistake? And as experiments shown (fig 1), the best choice of behavior policy is the most recent policy, which is obvious for on-policy methods and most researchers use this situation as default. Overall, I think this paper needs more effort to make the motivation.

---

> ### Author Response · Authors · 2022-08-01
> **Response to Reviewer GBX9**
>
> We are very grateful to the reviewer for their insightful review.
>
> You are correct that gradient accumulation can be used to reduce GPU memory requirements. However, we think that this is typically not as attractive as the approach we propose. This is because of the important distinction we draw between the optimization batch size (the number of environment steps in each gradient step) and the iteration batch size (the number of environment steps in each alternation between sampling and optimization:
>
> - Gradient accumulation can be used to achieve optimization batch size-invariance, but our approach of adjusting the learning rate or Adam step size is much simpler to implement and is likely to perform better when above the critical batch size.
> - Gradient accumulation does not help at all to achieve iteration batch size-invariance, which is what the remainder of our approach achieves. The iteration batch size may have its own constraints associated with it, such as CPU memory and iteration wall time.
>
> Our approach allows the degree of data parallelism (which affects both the optimization and the iteration batch sizes) to be freely adjusted according to computational constraints, while other hyperparameters are adjusted formulaically to compensate. The introduction and the discussion in Section 6.2 have been updated to better convey this motivation.
>
> You are also correct that it is best for the behavior policy to be as recent as possible, in the sense that staleness hurts performance. However, it is important to distinguish between the behavior policy and the proximal policy, and our experiments show that it is *not* always best for the proximal policy to be as recent as possible. For example, in our ablation (c) (Figures 2 and 3), failing to adjust the EWMA leads to a proximal policy that is too recent, which hurts performance. Further study of the effect of the age of the proximal policy is given in Appendix G. Based on the analysis and experiments given there, we believe that using a more recent proximal policy effectively lowers the strength of the KL penalty, and that adjusting the KL penalty coefficient to compensate for this effect is insufficient because there is still an overall effect of increasing the noise of the KL penalty.

---

> > ### Comment · Reviewer_GBX9 · 2022-08-07
> > **Thanks for responses**
> >
> > Thanks for the authors' responses. I agree that the two batch invariants should be separately analyzed, my main concern is that the second batch invariants lack some theoretical guarantees so the whole method is not particularly appealing to me. I think it would be interesting to further explore how old policies are suitable for updating and how can adjust the proximal policies to improve performance, which is a bit different from keeping batch size-invariance property. And it might be better if gradient accumulation can be used as a baseline.  I'll keep my score at a 4.

---

### Official Review · Reviewer_vc65 · 2022-07-06

**Rating:** 7
**Confidence:** 4
**Soundness:** 4 excellent
**Presentation:** 4 excellent
**Contribution:** 3 good

**Summary:**

This paper investigates how to make on-policy surrogate-objective-based algorithms PPO and PPG batch-size invariant: adjustments to their batch size can be compensated for by automatic adjustments to other hyperparameters. To enable this, they discuss a different between two uses of the old policy in the calculation of the PPO/PPG objective, and propose a decoupling of these uses. This decoupling then allows them to propose variants of PPO and PPG that are batch-size invariant, and they demonstrate across the procgen task suite that their adjustments mostly produce batch-size invariant results. They show their variants also outperform the standard versions of PPO and PPG.

**Questions:**

L171: what exactly do you mean here? What hyperparameter is being set to the output of this calculation?

**Limitations:**

I think the limitations are well-discussed.

**Strengths And Weaknesses:**

The paper is well-written and clearly describes batch size invariance, the decoupling of behaviour and proximal policy in the PPO objective, and the adjustments to PPO and PPG to make them batch-size invariance. The experimental results back up their claims that the adjustments do make PPO/PPG batch-size invariant, and the fact that their adjustment improves these algorithms is a nice added benefit. The contribution is novel and of some use to the community.

I think the main weakness is that the desire to have these algorithms batch-size invariant isn't as well-motivated as it could be. There is some discussion in the introduction, but it would be useful to discuss practical cases where batch-size invariance would be beneficial. Further, the discussion that there is (L32-35) motivates that increasing the effective batch size (to reduce variance) through these methods would be useful, but there aren't any experiments showing that these adjustments can be used to increase the batch size to actually reduce the gradient variance, and so an improvement to the paper would be to formulate those kinds of experiments.

Also, given you're evaluating on Procgen already, it would be great to see generalisation experiments - training on the standard subset of levels and reporting the evaluation score on the unseen levels. I'd expect the results to look similar to the current ones (i.e. that batch size invariance is maintained even when testing for generalisation performance), but it would be great to empirically validate that.

EDIT: I've read the rebuttal and feel it's addressed my concerns, but I'll keep my score at a 7 as I think that's the level of the paper.

---

> ### Author Response · Authors · 2022-08-01
> **Response to Reviewer vc65**
>
> We are very grateful to the reviewer for their thoughtful review.
>
> We agree that batch size-invariance could have been better motivated. The essential motivation is that the batch size has a big influence on training, but is often constrained by computational resources such as GPU memory. If an algorithm is batch size-invariant, then the batch size may be freely adjusted according to computational constraints, while other hyperparameters are adjusted formulaically to compensate. The introduction and the discussion in Section 6.2 have been updated to better convey this motivation.
>
> Regarding the question about line 171, \beta_{prox} should be modified such that the given expression is multiplied by c. This sentence has been updated to make it clearer.

---

> > ### Comment · Reviewer_vc65 · 2022-08-05
> > **Response**
> >
> > I've read the rebuttal and feel it's addressed my concerns, but I'll keep my score at a 7 as I think that's the level of the paper.

---

### Official Review · Reviewer_SfJ4 · 2022-07-07

**Rating:** 6
**Confidence:** 3
**Soundness:** 3 good
**Presentation:** 3 good
**Contribution:** 2 fair

**Summary:**

The authors propose an approach to make policy optimization algorithms such as PPO batch size invariant, where essentially performance achieved by the algorithm can be preserved even after changing batch sizes by changing relevant hyperparameters accordingly.

Thanks to the authors for putting in the effort in doing this work!

**Questions:**

Questions:
- I am not clear why batch size invariance would be a property we want. Is it that larger batches are preferred due lower variance in gradient updates (as mentioned in the paper), but that is not possible due to GPU load, so ideally we want an algorithm where we can have a smaller batch size but achieve performance as though we were using a larger batch size?
- On line 76, the paper mentions another work that proposes the similar objective, but the paper doesnt make any comparisons. I am curious how their modifications relate to yours and why theirs could not be used here?
- In Figure 1(c), the paper says “holds back learning”. But I don’t see why this is true. Comparing to Figure 1(b), the performances in 1(c) don’t seem to be “held back”
- Maybe I am misunderstanding, but shouldn’t Figure 4 (right) be 4 sets of bar graphs: one for PPO-EWMA, PPO, PPG-EWMA, and PPG? Here there seem to be only two.

Suggestions:
- The paper lacks a dedicated literature review, which would be useful for context.
- As presented, the proposed improvement seems to be marginal (and at memory cost). I wonder if the benefit of this suggestion can be made clearer in a setting where the error due to coupled objective compounds and then introducing the decoupled objective can avoid that error compounding (this is a very abstract suggestion, but I am trying to see how the proposed suggestion can show much larger performance improvements). Are there actor critic algorithms that also do this type of coupling? I wonder if in those cases there is error in actor and critic estimations due to coupling, but by introducing this decoupling concept one could improve learning in both actor and critic, thus having a much better improvement over the vanilla version.
- Minor: FYI, the appendix was attached with the main paper.


**Limitations:**

Yes. This is more fundamental work with no direct societal impact.

**Strengths And Weaknesses:**

Strengths:
- The paper is well written and flows well.
- Labelling of the equations in Section 2 was very helpful.
- Figure 3 was helpful.
- I appreciate the acknowledgement in line 244 that the improvement is small with EWMA.

Weaknesses:
- The purpose of creating a batch size-invariant algorithm is not absolutely clear. It may be obvious to some folks but it wasn't to me. Line 35 attempts to give some explanation regarding data parallelism, but still was not absolutely clear (see questions below).
- Given the marginal improvement (as they’ve also acknowledged) and the additional memory cost (with EWMA) it's unclear to me that this would be readily adopted. I think it's a very interesting find, but it's unclear if it makes enough improvement that this would be used instead of vanilla PPO (see suggestions below).

---

> ### Author Response · Authors · 2022-08-01
> **Response to Reviewer SfJ4**
>
> We are very grateful to the reviewer for their detailed review.
>
> You are correct about the motivation for batch size-invariance. Put another way, the batch size has a big influence on training, but is often constrained by computational resources such as GPU memory. If an algorithm is batch size-invariant, then the batch size may be freely adjusted according to computational constraints, while other hyperparameters are adjusted formulaically to compensate. The introduction and the discussion in Section 6.2 have been updated to better convey this motivation.
>
> Regarding Mirror Descent Policy Optimization (MDPO), they also mention that their objective can be decoupled, but the focus of that work is on other differences between PPO and MDPO, such as the use of minibatches and the direction of the KL penalty. We did not compare our method with theirs because we were focused on the effect of decoupling rather than on the effect of these other differences. In a sense, our work studies the same modification that they mention, but applied to PPO rather than MDPO, and in greater depth.
>
> Regarding Figure 1(c), learning is "held back" for a staleness of 1, 2, 4 or 8 iterations, compared to figure 1(b). We explain in the caption that this is not the case for a staleness of 16 or 32 iterations, where we say that "additional stability is helpful for very stale data".
>
> Regarding Figure 4, the right-hand plot shows the difference in normalized return between the EWMA and the non-EWMA algorithms, i.e. PPO-EWMA performance minus PPO performance and PPG-EWMA performance minus PPO performance. We show this to isolate the effect of the EWMA. The caption for this figure has been updated to better explain this.
>
> Regarding the literature review, there was previously a literature review on batch size-invariance in Appendix C, but this has now been expanded to include a review of related work on policy optimization, including MDPO, and moved to the main body of the paper.
>
> The suggestion about compounding error and critic coupling is interesting, but as far as we are aware, learning algorithms for critics typically do not involve a proximal policy, and so do not involve any coupling.

---

> > ### Comment · Reviewer_SfJ4 · 2022-08-05
> > **response to authors**
> >
> > Thank you authors for your response. It addressed by questions. I will keep the score as is.

---

### Official Review · Reviewer_PXCQ · 2022-07-16

**Rating:** 7
**Confidence:** 4
**Soundness:** 3 good
**Presentation:** 4 excellent
**Contribution:** 4 excellent

**Summary:**

#

This paper studies how to achieve batch-invariance in the policy optimization methods, e.g., PPO. Changing batch-size (the samples an on-policy algorithm collects per update) has a significant influence on policy learning for existing algorithms; batch-invariance will help to increase the batch size with limited computational resources.

In vanilla gradient descent, one can achieve batch invariance by adjusting learning rates, but the learning speed of the policy is affected by both the learning rate and the KL constraints. The paper first distinguishes the behavior policy and the proximal policy and observes that the two do not need to be equal. The reason is that though we need to use the exact behavior policy that collects the data to compute the objective for a good gradient estimation, its age does not matter too much. In contrast, we do not need to use the proximal policy in the KL constraint to sample data, but its age is directly related to how fast the policy can change.

The paper designs a decoupled clipped objective. Such decoupling allows the authors to use an exponentially-weighted moving average (EWMA) of the policy network as the proximal policy to maintain invariance when the algorithm has different policy update sizes. With additional batch-invariance techniques in the gradient descent, the method can maintain similar performance when the batch size changes.

**Questions:**

see above

**Limitations:**

yes

**Strengths And Weaknesses:**

Actually I like this paper.

- Batch size has huge influence on policy learning performance but is not emphasized in many publications. I especially appreciate the advice in 6.2. I have seen many beginners ignore its importance and struggle to tune the parameters in PPO. The comprehensive study of the batch size and the practical advice would be excellent guidance for them to understand the optimization process. I would recommend my colleague to read the paper when it is published.
- The decoupled objective is novel to me. It provides a new perspective on the constraints and objectives in PPO and could help us understand the dynamics of policy optimization. It reminds me of the distributional RL, where we can train multiple agents but have to use a centralized policy to constrain their differences. I wonder if there is any connection between the two.
- I think this paper provides a comprehensive recipe to make PPO batch-invariant. The modifications is very practical and perform good in experiments. The technique contribution is solid. It is a pity that EWMA does not have too much effects.

As for the weakness:

- I only found experiments on procgen. I wonder how it will behave on continuous control tasks, where the original PPO paper did its experiments. I am also curious about if PPG will differ from PPO in those domain.

---

> ### Author Response · Authors · 2022-08-01
> **Response to Reviewer PXCQ**
>
> We are very grateful to the reviewer for their generous review.
>
> Unfortunately we did not run any experiments on continuous control environments. We found  Procgen Benchmark to serve as a good testbed for our methods due to the difficulty and diversity of the environments. We hope to see experimental results on other environments in future work.

---

> > ### Comment · Reviewer_PXCQ · 2022-08-07
> > **Response**
> >
> > Thank you for your response. I will keep the score as is.

---

### Meta-Review · Area_Chair_QCVK · 2022-08-26

**Recommendation:** Accept
**Confidence:** Less certain

**Metareview:**

The paper provides a practical approach for making policy optimization methods (e.g., PPO) batch-size invariant. Batch-size invariance allows for achieving the same algorithmic behaviour when different computational resources are available (here the trade-off is the batch size). Almost all the reviewers consider the paper interesting and sound. The paper may be of large interest to practitioners and it may allow a much simpler scaling/reproduction of existing algorithms/experiments.

**Award:**

No

---

### Decision · Program_Chairs · 2022-09-14

Accept